

# Impact of self-efficacy and burnout on professional development of physical education teachers in the digital age: a systematic review

Luhong Ma[1,*], Chen Soon Chee[1], Saidon Amri[1], Xuejiao Gao[1], Qinglei Wang[2], Nina Wang[3] and Pan Liu[1,*]

[1] Department of Sports Studies, Faculty of Educational Studies, Universiti Putra Malaysia, Seri Kembangan, Serdang, Malaysia
[2] Faculty of Sports and Exercise Science, Universiti Malaya, Kuala Lumpur, Malaysia
[3] Department of Educational Psychology & Counselling, Faculty of Education, Universiti Malaya, Kuala Lumpur, Malaysia
* These authors contributed equally to this work.

Corresponding authors
Chen Soon Chee,
cschee@upm.edu.my
Pan Liu, 498836060@qq.com

## ABSTRACT

**Background:** The professional development of teachers in the digital age will positively impact the effectiveness of physical education teaching. Exploring key factors such as self-efficacy, burnout, and digital technology is crucial to ensure the professional development of teachers.
**Methods:** The search was conducted in accordance with the PRISMA guidelines and utilized the following databases: Scopus, Web of Science, ProQuest, and Google Scholar. Inclusion and exclusion criteria: population, research methods, keywords, and time limit were described for this study. This article predominantly includes cross-sectional studies, so we have used the AXIS risk assessment methodology.
**Results:** The study included ten articles, seven of which (70%) were quantitative. Three key findings emerged from this review: first, the studies on self-efficacy were more noteworthy than the studies on burnout. Second, female teachers were more expressive in their digital teaching, while male teachers had higher levels of self-efficacy in their digital teaching. Finally, the study explored various factors affecting self-efficacy and burnout in relation to digital teaching. The study demonstrated that professional development has a higher impact on physical education teachers' self-efficacy, and in turn, self-efficacy reduces burnout. Additionally, burnout had a significant impact on professional development.
**Conclusion:** This study describes the limitations of risk assessment and uses the AXIS tool to assess the methodological quality of this review report instead of using the risk of bias tool. The use of digital teaching methods can increase self-efficacy and alleviate burnout among physical education teachers. This review analyses the effects of digital technology, self-efficacy, and burnout on the career progression of physical education instructors and examines the implications for future developments.

## INTRODUCTION

The progress made in the digital technology used in physical education has been remarkable, providing a wide range of resources and tools for physical education teaching (*Khitskov et al., 2017*). It has introduced concepts such as flipped classrooms, massive open online courses (MOOCs), TikTok, live streaming, and recorded courses. These technological innovations have been applied in the field of physical education, seamlessly integrating online (*Jiang et al., 2023*) and face-to-face teaching (*Daum, 2012*). Moreover, educators employ apps and other technological resources (*Zhou, 2021a*) to oversee students' physical well-being, delegate assignments, and facilitate feedback and communication. Consequently, the primary technological underpinning of this overview is the blended physical education model (*Wang et al., 2022*; *Zhou, 2021b*). It is a teaching method that uses a combination of face-to-face and online instruction (*Kastrena, Setiawan & Adawiyah, 2020*), which is not only a practical outcome of the digital age, but has become a prominent approach brought about by the COVID-19 pandemic. Under the current educational model, it is essential to explore the effects of self-efficacy and burnout on teachers' professional development. It can provide a theoretical basis for a rational and effective professional development programme with academic administrators and policymakers. The aim is to promote teachers' professional development, reduce burnout and maximise the effect of self-efficacy.

### Definition of relevant variables

In digital physical education, teachers need to continuously improve their digital professional development (*Khitskov et al., 2017*; *Olesov et al., 2020*). Professional development is more specifically defined among teachers, with *Bates & Morgan (2018)* emphasizing that professional development is a pedagogical activity whose aim is primarily to improve teachers' knowledge and skills. It is worth noting that teachers are required to learn and adapt to new teaching tools and technologies. Hence, the professional development of physical education teachers in this study pertains primarily to their acquisition of professional knowledge and skills for adapting to and using digital technologies. However, teachers may face the double pressure of learning and working while dealing with these high-tech technologies. This in turn can lead to burnout. Burnout, a concept introduced by *Freudenberger (1974)* and later refined by *Maslach, Schaufeli & Leiter (2001)*, encompasses emotional exhaustion, depersonalization, and personal achievement. Teacher burnout is affected by intrinsic factors (*e.g.*, age, gender, self-efficacy, and motivation) and extrinsic factors (*e.g.*, organizational climate, satisfaction, and working conditions) (*de Carvalho Silva Pereira, Holanda Ramos & Leal Soares Ramos, 2020*). Self-efficacy is an individual's evaluation and confidence in performing a particular task (*Bandura, 1993*). In this study, self-efficacy primarily refers to the specific beliefs that emerge as physical education teachers engage with digital technology. When exploring the factors leading to burnout, there was an emphasis on how self-efficacy not only alleviates burnout levels, but also contributes to the professional development and growth of
teachers, thus reinforcing their beliefs about teaching and learning (*Barnes et al., 2018*; *Bümen, 2010*; *Sang et al., 2022*). Exploring how burnout and self-efficacy affect professional development and the relationship between them is the focus problem of this study. Therefore, it is necessary to conduct a systematic review of the impact of burnout and self-efficacy on the professional development of physical education teachers.

## Burnout and professional development

Burnout is influenced by professional development (*Heppe et al., 2024*; *Roberts et al., 2020*; *Smith & Smalley, 2018*). During teachers' professional development, burnout is a psychological issue that teachers face. Based on the hierarchical regression model (HRM) and qualitative comparative analysis (QCA), the main factors that contribute to burnout and physical and mental health problems are the availability and quality of resources, information, workload, lack of organizational justice, interpersonal conflict and job insecurity. Self-efficacy is also a factor (*Gómez-Domínguez et al., 2022*). The more advanced the teachers' competence and attitudes towards work, the lower the teacher burnout (*Koczon-Zurek, 2007*). Relevant studies have also shown a negative correlation between burnout and the willingness to succeed professionally among teachers, which causes them to quit their jobs, or continue working without investing in professional development (*Fiorilli et al., 2015*). Studies have shown that a teacher's technology integration, self-efficacy and professional competence is influenced by gender and age (*Şen & Yildiz Durak, 2022*). In general terms, burnout can affect and threaten teachers' work, as well as their attitudes toward professional development (*Cacciamani et al., 2022*). Burnout can negatively impact teachers' professional development and pedagogical practices (*Ezza Mad et al., 2022*), thus affecting the quality of physical education programs, classroom management, and students' interest in learning (*Barnes et al., 2018*). Hence, there is a strong relationship between burnout and professional development.

## Digital self-efficacy and professional development

Integrated digital technologies in physical education instruction present both opportunities and challenges for educators. Teacher self-efficacy is defined as the teacher's belief in his or her ability to accomplish teaching tasks and to develop professionally and effectively (*Kavgacı, 2022*). Teachers must constantly increase their competence in digital professional development, while also being aware of their burnout and self-efficacy levels (*Knopik & Domagała-Zyśk, 2022*). By creating an optimal working environment and offering comprehensive support and training opportunities (*Martínez-Rico et al., 2021*; *Mezzaroba, Zoboli & Moraes, 2019*), teachers can more effectively manage the challenges of physical education instruction in the digital age, improve teaching standards, and increase pupils' engagement and teaching outcomes (*Ma & Zhou, 2021*). *Brown, Powers & Olden (2020)*, on the Concerns about Adoption Model (CBAM), researchers found a significant effect between teachers' professional development and self-efficacy. Teachers need to be confident enough to design and implement online or hybrid courses.

Understanding the impact of burnout and self-efficacy on the professional development of physical education teachers during the digitization of physical education instruction can provide new insights for future professional development training.

## Digital self-efficacy, burnout and professional development

It is worth noting that decreasing teacher burnout proves more advantageous in teaching interventions than improving self-efficacy, which stimulates the need for teacher professional development (*Iaochite & de Souza Neto, 2014*). Researchers have shown that self-efficacy is a valid predictor of teachers' job satisfaction and well-being (*Caprara et al., 2006*; *Cooper, 2019*). Teachers with higher self-efficacy are more likely to uphold their work ethic, job satisfaction, and well-being, which decreases the rate of burnout (*Geraci et al., 2023*). In other words, self-efficacy shows negative predictability for burnout (*Naz, Atta & Malik, 2017*). Self-efficacy was negatively related to burnout (*Bottiani et al., 2019*; *Fathi & Derakhshan, 2021*; *Zee & Koomen, 2016*). There is a significant positive correlation between self-efficacy and teacher professional development (*Li, Manoharan & Cui, 2022*; *Lumpe et al., 2014*; *Martin et al., 2009*; *Salari & Farahian, 2022*). Conversely, and unlike self-efficacy, teacher professional development is negatively correlated with burnout (*Özer & Beycioglu, 2010*). The results found that teachers who experienced professional frustration and burnout had higher and more positive attitudes toward growth in their professional development. Thus, the burnout phenomenon may be short-lived (*Chang, 2009*). Positive correlations between teachers' professional development and teaching self-efficacy were both direct and indirect negative predictors of teacher burnout (*Lauermann & König, 2016*). Symptoms of burnout are influenced by the extent to which teachers experience professional growth, self-efficacy, and the environment of perceived success in professional development, which decreases (*Bümen, 2010*).

Performing a systematic review on burnout and self-efficacy in the digital age, specifically focusing on their impact on the professional development of physical education teachers, is crucial for enhancing our understanding of physical education teacher development in the digital era. While there is extensive research on self-efficacy and burnout in education as a whole (*Barni, Danioni & Benevene, 2019*; *Holly, 2020*), the literature review concerning burnout and self-efficacy in the specific field of physical education (*Feltz, Short & Sullivan, 2008*) is relatively limited. This study aims to analyze the current status of burnout, self-efficacy, and professional development in digital technology for physical education teachers by synthesizing the available literature and proposing future research directions. The primary research questions of this study are as follows:

1. What are the constraints to professional development of PE teachers?

2. How do digital tools in physical education instruction contribute to teacher development in the digital age?

3. How do self-efficacy and burnout influence the professional development of physical education teachers?

4. What potential future pathways exist for the professional development of physical education teachers?

## MATERIALS AND METHODS

### Eligibility criteria

The inclusion criteria of the literature for this study were: a) it had to focus on physical education (PE) teachers as the main population; b) the research methods had to involve digital, information and communications technology (ICT), and blended learning models; c) the articles had to explore self-efficacy, burnout, or professional development, or a combination of these; d) the literature had to have been published in English; e) the earliest reference to flipped classrooms had to be from the USA in 2007. The reason for the last criterion was because flipped classrooms were pioneered by American educators Jon Bergmann and Aaron Sams (*Hammer, 2007*). The concept centers around leveraging technology to enhance self-directed learning and plays a pivotal role in shaping the landscape of digital classrooms (*Akçayır & Akçayır, 2018*; *Bishop & Verleger, 2013*). Digital technologies were rapidly adopted and developed during 2019–2023, the reason for this is the outbreak of the epidemic and the shift to online teaching. So the most appropriate years are between 1$^{st}$ January 2007 and 31st July 2023.

Exclusion criteria were: a) non-physical education teachers, preschool teachers, and playschool teachers; b) book chapter reviews, conference abstracts, and reports; c) articles not published in English; d) literature that was not available in the full text and abstracts.

### Search strategy

This review conforms to PRISMA 2020, the preferred reporting standard for systematic reviews and meta-analyses. It followed the guidelines of this standard for data collection, selection, and analysis (*Page et al., 2021*). This review is registered with the INPLASY database (https://inplasy.com/) under the registration number INPLASY202360026 and DOI number 10.37766/inplasy2023.6.0026. This study was conducted in 31st July 2023 by searching English databases. These databases were chosen for the literature review due to their comprehensive coverage across disciplines (Web of Science and Scopus), specialized focus on dissertations and theses (ProQuest), and the ability to capture diverse sources, including real-time information (Google). This ensured a thorough and up-to-date examination of the chosen topic. For each search, topics and abstracts were selected using the following terms as search criteria: "self-efficacy" OR "burnout" OR "self-efficacy and burnout" OR "burnout and professional development" OR "self-efficacy and professional development", "online" OR "ICT" OR "digital" OR "internet" OR "online learning" and "physical education teacher" OR "physical education teacher".

### Study selection

The literature screening process involved two authors, L.H.M. and P.L., who independently selected and adhered to the inclusion criteria for selection. First, the authors searched the literature and reviewed abstracts to compare and summarize the data. In the second step, the summarized literature was imported into EndNote, removing duplicates and irrelevant information. Subsequently, the literature initially selected for inclusion was

brought into the Rayyon tool (*Ouzzani et al., 2016*), a systematic review platform designed to enhance the efficiency and management of the literature review and systematic review processes. The full text was read independently by the first (L.H.M.) and second (P.L.) authors to screen for the inclusion/exclusion criteria. Studies marked as eligible by both authors were directly included. Additionally, in the event of inconsistent results, the third author (X.J.G) was consulted to reach a consensus before deciding to include or exclude the literature. This was done by two authors writing the reason for the challenge in the Rayyon tool, and then a third author reading the full text and giving his or her opinion. If two out of the three authors were in agreement, the final decision was determined by a majority vote. Then, Q.L.W. and N.N.W reviewed the three authors' opinions. Finally, C.S. C. and S.A. reviewed and checked the data as well as the entire article.

## Data extraction

We started by identifying and extracting data, which consisted of six key elements: (a) publication information (author and year), (b) sample data information (country, number, and category), (c) digital tools (computer technology, online teaching, blended teaching, flipped classroom, distance learning, *etc*.), (d) research methodology, (e) research area (self-efficacy, burnout, professional development, and other related aspects), and (f) main findings and discussion (Table 1). The extracted information is tabulated and summarised in categories.

## Quality assessment

The methodological quality assessment in this study primarily employed a cross-sectional study design utilizing the AXIS tool, a widely used instrument across various research disciplines (*Parisi et al., 2020*; *Pellicane & Ciesla, 2022*; *Robson, Allen & Howard, 2020*; *Rodriguez-Morales et al., 2020*). The AXIS tool assesses the internal and external validity relating to five main areas of study, namely introduction, research methodology, results, discussion, and others. These areas included 20 items used in the evaluations (*Downes et al., 2016*). This study is primarily concerned with cross-sectional studies, and the AXIS tool is the best for evaluating cross-sectional studies. For this reason, the researchers used the AXIS tool to assess the methodological quality of this review report instead of using the risk of bias tool. To aid in the evaluation (Table 2), the AXIS tool was categorized by the researchers into a total of three assessment outcomes (YES = Y, NO = N, DON'T KNOW = C), and the scored values were transformed. Each "Yes" was marked as a "1", each "N" was marked as a "0", and each "C" was marked as a "0.5" (*Rovito et al., 2021*). Finally, the scores were totaled and their percentages were used to determine the quality assessment scale. Scores of ≥70% were considered excellent quality, scores between 50–69% were considered medium quality, and scores below 50% were considered low quality (*Rovito et al., 2021*; *Scott & Pocock, 2021*). The first author completed the assessment, and the responsibility for reviewing the AXIS results for bias was given to the last two authors, N.N.W. and Q.L.W.

**Table 1 Summary of the key extracts from the project.** The authors, year, country, sample size, educational category, tools digitised, research methodology and field, main findings and discussion of the ten articles.

| Title and Author | Year | Country | Sample | Type of education | Tools | Methodology | Research field | Key findings | Discussion |
|---|---|---|---|---|---|---|---|---|---|
| Remote Teaching, Self-Resilience, Stress, Professional Efficacy, and Subjective Health among Israeli PE Teachers during the COVID-19 Pandemic (Ben Amotz et al.) | 2022 | Israel | 757 | K12 | Remote teaching | QT (SE, Resilience, Subjective wellbeing) | SE, stress/health perception levels. | Higher competence and effectiveness of female teachers in distance learning | Increased training for PD |
| Primary School Physical Education at the Time of the COVID-19 Pandemic: Could Online Teaching Undermine Teachers' Self-Efficacy and Work Engagement? (Gobbi et al.) | 2021 | Italy | 622 | Primary | Online teaching | QT (SE, PDC) | SE, work engagement, Digital PD ability. | 1. Low digital competence and SE reduced SE. 2. SE is related to work engagement. | Improve digital PD training opportunities. |
| The Relationship between attitudes Of prospective physical education teachers towards education technologies and computer self-efficacy beliefs (Kalemoğlu Varol) | 2014 | Turkey | 377 | University | computer technology | QT (TA, CSEB) | CSEB, TA, PD | Male teachers had higher computer skills and self-beliefs than female teachers. | SE increases the more computer-based educational technology is used. |
| Teachers' and students' perceptions of factors influencing the adoption of information and communications technology in physical education in Singapore schools (Koh et al.) | 2022 | Singapore | 11 | K12, University | ICT, flipped learning; | QL (semi-structured interview) | SE(ICT), | SE is influenced by ICT capabilities. | Provides PD opportunities that increase teachers' confidence in "integrating ICT PE skills". |
| Implementing the adapted physical education E-learning program into physical education teacher education program (Kwon & Block) | 2017 | Korea | 74 | University | e-learning | RCT (APE plan) | SE | Improved SE of pre-service teachers. | Increase PD opportunities and develop online resources. |
| Changes in physical education teachers' motivations predict the evolution of behaviors promoting students' physical activity during the COVID-19 lockdown (Maltagliati et al.) | 2021 | Italy France | 1,931 | / | Digital technology | QT (IPPA, BPPA, SDM, SE, Engagement) | SE, autonomous, controlled motivation, PUTDT, DA, PA, pressures | 1. SE was positively correlated with PUTDT, followed by motivation, control, and IPPA; 2. SE, PUTDT, and IPPA were related to changes in BPPA. | Suggested that further research could be done through teachers' emotional relationships (stress, depression). |
| Professional Sports Trainers' Burnout in Fully Online and Blended Classes: Innovative Approaches in Physical Education and Sports Training (Nguyễn et al.) | 2022 | Vietnam | 60 | Personal gym trainers | Blended learning, fully online courses | M (burnout questionnaire, Focus-Group Interview) | Digital SE, burnout, | 1. Burnout levels tended to increase in fully online instruction. 2. Burnout increased; self-achievement decreased. | Increased PD opportunities for coaches, teachers and digital competence. |
| Self-evaluated teacher effectiveness in physical education and sports during schools closedown and emergency distance learning: Teacher effectiveness in physical education (Şahin) | 2021 | Turkey | 172 | / | Distance education | QT (Teacher Effectiveness) | Digital SE, teaching technology | Learning environment, application of PE content, teaching strategies, and total score parameters were lower. | Support for future directions in research PD. |
| Digital burnout and perceptive failure: The example of physical education teachers during the COVID-19 pandemic (Şirin et al.) | 2022 | Turkey | 504 | High, secondary | Digital | QT (DB, PF) | Perceptual Fatigue, burnout | Moderate PF and DB. | Tuning in to PF and adapting to the digital environment. |
| Exploring HPE Teachers' Self-Efficacy Toward Technology Integration (Werner) | 2020 | USA | 57 | K12 | Computer technology | QT (CTISPE) | SE, PD | 1. PD increased computer SE. 2. No significant difference in age, or teaching experience. | PD using SE theory in conjunction with new technology. |

**Note:**

QT, Quantitative; QL, Qualitative; M, Mix method; SE, self-efficacy; PD, professional development; TA, Technology Attitude; CSEB, Computer SE Belief; PDC, perceived digital competence; ICT, Information and communications technology; PUTDT, perceived usefulness toward digital technologies; DA, decreased Amotivation; SDM, Self-determined motivation; CTISPE, Computer Technology Integration Survey for Physical Education; DB, digital burnout; PF, perceptual fatigue; PA, physical activity; PE, physical education; IPPA, Intention to promote PA; BPPA, behavior promote PA.

**Table 2 Methodological quality assessment by AXIS tools.** The AXIS tool includes five main areas of study, namely introduction, research methodology, results, discussion and others, which mainly involve 20 items evaluations. Yes for scale "1", No for scale "0", Don't know for scale "0.5", with scores of ≥70% being excellent quality, between 50–69% being medium quality, and below 50% being low quality.

**Introduction**

| Q1 | 1 Were the aims/objectives of the study clear? |

**Methods**

Q2    2 Was the study design appropriate for the stated aim(s)?

Q3    3 Was the sample size justified?

Q4    4 Was the target/reference population clearly defined? (Is it clear who the research was about?)

Q5    5 Was the sample frame taken from an appropriate population base so that it closely represented the target/reference population under investigation?

Q6    6 Was the selection process likely to select subjects/participants that were representative of the target/reference population under investigation?

Q7    7 Were measures undertaken to address and categorise non-responders?

Q8    8 Were the risk factor and outcome variables measured appropriate to the aims of the study?

Q9    9 Were the risk factor and outcome variables measured correctly using instruments/measurements that had been trialled, piloted or published previously?

Q10    10 Is it clear what was used to determined statistical significance and/or precision estimates? (*e.g.*, *p* values, CIs)

Q11    11 Were the methods (including statistical methods) sufficiently described to enable them to be repeated?

**Results**

Q12    12 Were the basic data adequately described?

Q13    13 Does the response rate raise concerns about non-response bias?

Q14    14 If appropriate, was information about non-responders described?

Q15    15 Were the results internally consistent?

Q16    16 Were the results for the analyses described in the methods, presented?

**Discussion**

Q17    17 Were the authors' discussions and conclusions justified by the results?

Q18    18 Were the limitations of the study discussed?

**Other**

Q19    19 Were there any funding sources or conflicts of interest that may affect the authors' interpretation of the results?

Q20    20 Was ethical approval or consent of participants attained?

**Results**

| Yes | No | Don't know/comment |
|---|---|---|
| | 0 | 0.5 |
| 1 | | |

| Score | Excellent ≥70%, medium 50–69%, low <50% |

# RESULTS

The section outlines the research methodology and presents the current findings. In the extraction data process, based on the main research questions of this study: burnout among college physical education teachers, how self-efficacy affects professional development and the relationship between them, combined with the six concerns of the research plan (data extraction), inclusion criteria, and exclusion criteria. Then, this section objectively describes the fundamental aspects of professional development, self-efficacy, and burnout among physical education teachers in the digital age.

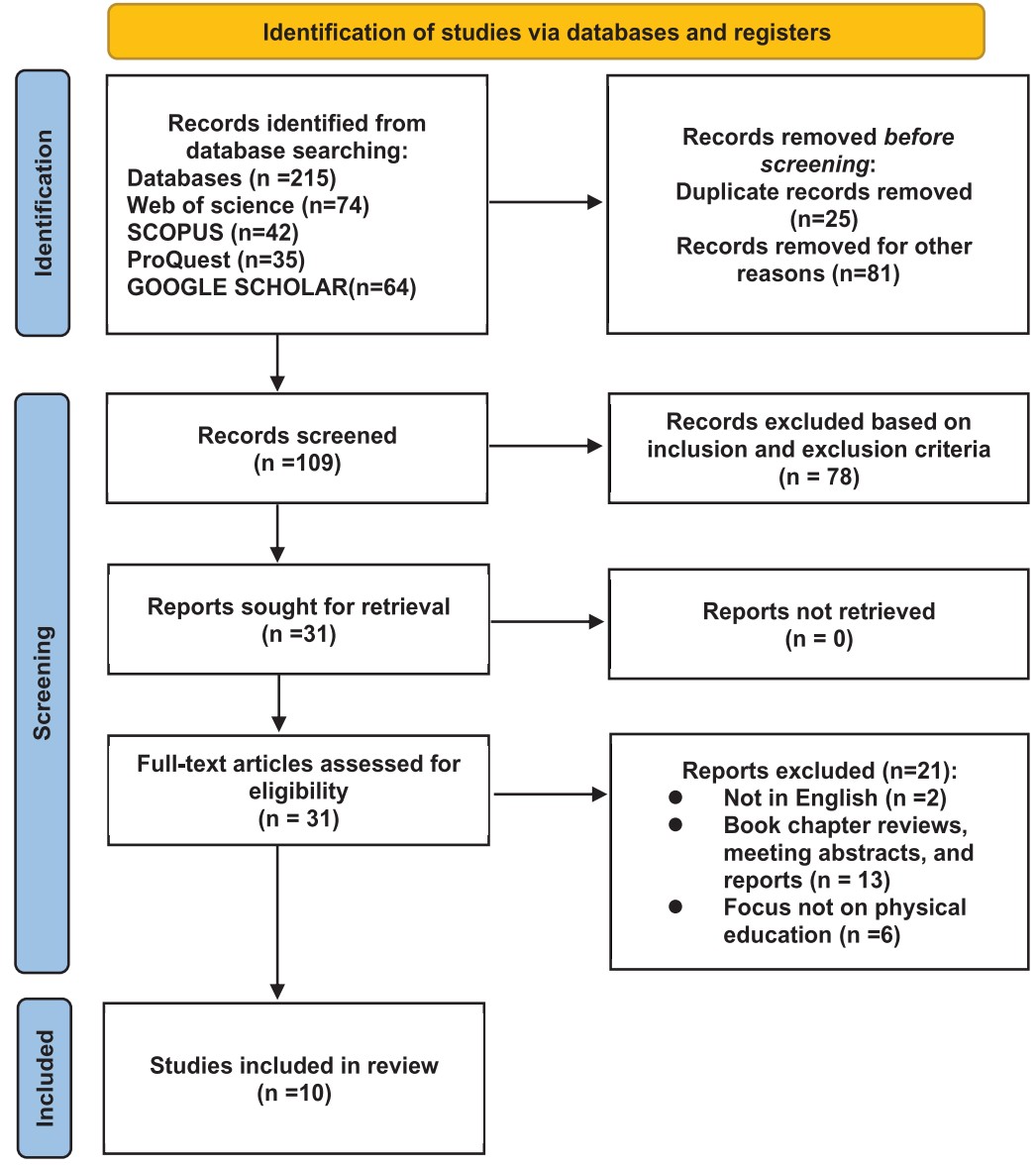

**Figure 1 Flowchart for identifying, screening and incorporating data based on Prisma.** In the Identification step, a total of 215 articles were considered. In the Screening step, 31 articles were selected based on the inclusion and exclusion criteria. Finally, in the included step, the full text of the selected articles was read, leading to the inclusion of ten articles.

## Selection process results

The researchers searched four databases, including Web of Science (74), Google (64), ProQuest (35), and Scopus (42), which resulted in a total of 215 articles. The preliminary review removed duplicates and literature that was not full text, resulting in a total of 109 articles. The literature was then imported into Endnote software to create a full-text database of article titles, abstracts, and author information. After a secondary screening using the inclusion and exclusion criteria, 78 irrelevant articles were eliminated, resulting in a total of 31 articles. Finally, the remaining 31 literatures were imported into the Rayyan

Table 3 **Results of the AXIS tool review.** The methodological quality of ten publications was reported based on the 20 items of the AXIS tool. Y stands for yes, N stands for no, and UC stands for unclear or not known.

| Article number | Items | Introduction | Methods | | | | | | | | | | Results | | | | | Discussion | | Other | |
|---|---|---|---|---|---|---|---|---|---|---|---|---|---|---|---|---|---|---|---|---|---|
| | Author(s) year of publication | Q1 | Q2 | Q3 | Q4 | Q5 | Q6 | Q7 | Q8 | Q9 | Q10 | Q11 | Q12 | Q13 | Q14 | Q15 | Q16 | Q17 | Q18 | Q19 | Q20 |
| 1 | Ben Amotz et al. (2022) (12) | Y | Y | Y | Y | Y | Y | Y | Y | Y | Y | Y | Y | UC | N | Y | Y | Y | Y | Y | Y |
| 2 | Gobbi et al. (2021) (13) | Y | Y | Y | Y | Y | Y | Y | Y | Y | Y | Y | Y | N | N | Y | Y | Y | Y | Y | Y |
| 3 | Kalemoğlu Varol (2014) (2) | Y | Y | Y | Y | Y | Y | N | Y | Y | Y | UC | Y | N | N | Y | Y | Y | N | N | N |
| 4 | Koh et al. (2022) (1) | Y | Y | Y | Y | Y | Y | Y | Y | Y | Y | Y | Y | N | N | Y | Y | Y | Y | Y | Y |
| 5 | Kwon & Block (2017) (0) | Y | Y | Y | Y | Y | Y | Y | Y | Y | Y | Y | Y | N | N | Y | Y | Y | N | N | Y |
| 6 | Maltagliati et al. (2021) (0) | Y | Y | Y | Y | Y | Y | N | Y | Y | Y | Y | Y | N | N | Y | Y | Y | Y | N | Y |
| 7 | Nguyễn, Đỗ & Nguyễn (2022) (0) | Y | Y | Y | Y | Y | Y | N | Y | Y | Y | Y | Y | N | N | Y | Y | Y | Y | Y | Y |
| 8 | Şahin (2021) (2) | Y | Y | Y | Y | Y | N | N | Y | Y | Y | Y | Y | N | N | Y | Y | Y | Y | N | Y |
| 9 | Şirin et al. (2022) (1) | Y | Y | Y | Y | Y | Y | N | Y | Y | Y | Y | Y | N | N | Y | Y | Y | N | Y | N |
| 10 | Werner (2020) (0) | Y | Y | Y | Y | Y | Y | Y | Y | Y | Y | Y | Y | N | Y | Y | Y | Y | Y | N | N |

software for collaborative work, and each of the two authors re-read the full text with the digital age in mind. Two authors read the markup and removed 21 articles. Reference exclusion criteria include articles not in English, book chapters, conference abstract articles and reports, and research subjects that are not sports. A total of 10 studies were included (seven quantitative studies, one randomized controlled trial (RCT), one qualitative study, and one mixed study) (Fig. 1).

## Data extraction and risk assessment results

Based on the methodological data extraction design, Table 1 shows the primary focus of the remaining ten studies. The table not only gives the title, author information and year, but also extracts the primary digitisation techniques, the main findings and discussion. Examples of key findings are given below: physical education teachers' self-efficacy and burnout in the digital age have an impact on professional development. Conversely, professional development can significantly increase the level of digital self-efficacy for teachers. Burnout focuses on stress, motivation, competence, and mood in a teacher's professional development (Maltagliati et al., 2021). Self-efficacy is a main focus of current studies and a dominant research trend in the digital age. In contrast, burnout has been studied relatively infrequently, with only two studies in this review focusing on its impact on professional development (Nguyễn, Đỗ & Nguyễn, 2022; Şirin et al., 2022).

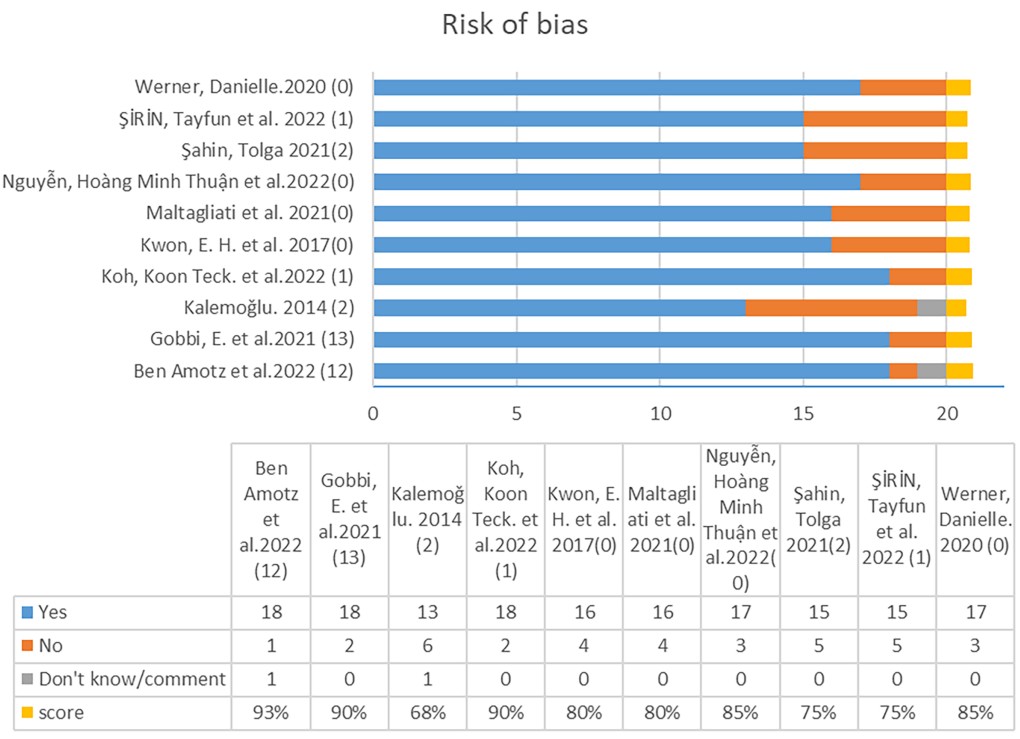

Figure 2 **Risk of bias assessment of the literature based on the AXIS tool.** The X-axis indicates the author and year of publication in the bar chart, and the Y-axis represents the score. Blue represents yes, orange represents No, grey represents Don't know, and yellow represents the final score.

   After conducting an independent review by two researchers, generating a query, and discussing with a third researcher, the following final review results were obtained (Table 3). Three articles scored above 0.9, four articles scored between 0.8 and 0.85, two scored below 0.75, and only one scored below 0.7 (Fig. 1). Based on the score percentages, nine articles reached a high quality rating, and one article was rated as upper medium quality, as shown in Fig. 2. Therefore, the quality of the literature review in this study is considered high.

## Analysis of digitalization of physical education trends

The inclusion criteria for this study began with the introduction of the flipped classroom in 2007 (*Hammer, 2007*). However, during the search and screening process, we found that there was very little literature dealing with digital self-efficacy, burnout, and professional development among physical education teachers that met the criteria. The final data included was from 2014 to 2022. There were three articles on digitalization for PE teachers published before 2020. There were seven projects in 2021–2022 (*Ben Amotz et al., 2022*; *Gobbi et al., 2021*; *Koh et al., 2022*; *Maltagliati et al., 2021*; *Nguyễn, Đỗ & Nguyễn, 2022*; *Şahin, 2021*; *Şirin et al., 2022*; *Werner, 2020*). This was mainly due to the outbreak of COVID-19, which forced most schools to close and suspend classes and change from face-to-face offline teaching to online teaching. The integration of digital technologies and

online instruction focused attention on the digital professional advancement of physical education teachers.

## Categories of participants

Nine out of the ten studies analyzed in this literature were studied in Asia and Europe ($n$ = 4,508), with only one participant survey conducted in the United States ($n$ = 57). The sample sizes ranged from 11 to 1,931, making a total of 4,565 participants. Besides, out of these items, five studies (*Ben Amotz et al., 2022*; *Gobbi et al., 2021*; *Koh et al., 2022*; *Şirin et al., 2022*; *Werner, 2020*) focused on physical education teachers in K12 education, middle school, and high school; three studies were conducted for college participants (*Kalemoğlu Varol, 2014*; *Koh et al., 2022*; *Kwon & Block, 2017*); one study mentioned individual personal teachers (*Nguyễn, Đỗ & Nguyễn, 2022*); and two reports did not specify the field levels of teaching (*Maltagliati et al., 2021*; *Şahin, 2021*). Gender also played a role in the findings, with female teachers showing more expressiveness during distance teaching (*Ben Amotz et al., 2022*) and male teachers demonstrating better computer skills and confidence (*Kalemoğlu Varol, 2014*). Overall, research has been conducted on physical education teachers at all levels of teaching, highlighting the significant role of digital development in physical education, with a particular emphasis on K12 PE teachers (*Ben Amotz et al., 2022*; *Koh et al., 2022*; *Werner, 2020*). In comparison, these aspects of physical education teaching at higher institutes of learning have not been studied as extensively. This could be due to the number of higher education institutions and the increased teaching demands in universities (*Maltagliati et al., 2021*).

## Digital tools and research

Teaching models in the digital era, specifically those embracing digitalized instructional methods (*e.g.*, online teaching, virtual reality (VR), augmented reality (AR), mobile apps, smart wearables, and digital curriculum design), are one of the most crucial indicators of components for studying the professional development competencies of physical education teachers. This review explores various digital teaching models and tools, including distance learning (*Ben Amotz et al., 2022*; *Şahin, 2021*), online teaching (*Gobbi et al., 2021*; *Kwon & Block, 2017*), computer technology (*Kalemoğlu Varol, 2014*; *Koh et al., 2022*; *Werner, 2020*), blended learning (*Nguyễn, Đỗ & Nguyễn, 2022*), flipped classrooms (*Koh et al., 2022*), and video conferencing (*Maltagliati et al., 2021*). The computer is the primary pedagogical tool used in digital teaching and learning. Consequently, computer competence is the primary indicator of a physical education teacher's digital professional development aptitude. As shown in a study by *Maltagliati et al. (2021)*, the perceived usefulness of digital technologies (b = 0.12, 95% CI [0.06–0.18], $p < 0.001$) was positively associated with behavioral change and intention to promote students' physical activity during physical education. The perception of the usefulness of digital technologies should be addressed in real-life teacher training (*Maltagliati et al., 2021*).The implementation of virtual reality (VR) and augmented reality (AR) technologies (*Nguyễn, Đỗ & Nguyễn, 2022*) can facilitate interactive and immersive experiences in physical education. In addition, integrating teaching and ICT tools is a necessary developmental tool for PE

teachers to use in digital teaching and learning (*Koh et al., 2022*). The exploration of qualitative methods could provide a more complete picture. According to the literature, "all prerequisites for integrating teaching and learning are professional development and the development of physical education teachers' ICT expertise skills for teaching and learning" (*Koh et al., 2022*). Another study showed that low digital competence of PE teachers led to lower student interaction and engagement (*Şahin, 2021*).

This review focuses on the statistics of research designs, with seven quantitative studies (*Ben Amotz et al., 2022*; *Gobbi et al., 2021*; *Kalemoğlu Varol, 2014*; *Maltagliati et al., 2021*; *Şahin, 2021*; *Şirin et al., 2022*; *Werner, 2020*), one qualitative study (*Koh et al., 2022*), one mixed study (*Nguyễn, Đỗ & Nguyễn, 2022*), and one RCT experiment (mainly involving pre-test and post-test data) (*Kwon & Block, 2017*). There were nine questionnaires involving self-efficacy and three on burnout, with only one examining both self-efficacy and the impact of burnout on professional development (*Nguyễn, Đỗ & Nguyễn, 2022*). Further research is required to explore the various aspects of digital self-efficacy, and studies incorporating burnout must be more comprehensive, extensive, and diverse. The focus on the digital professional development of physical education teachers is limited, perhaps because professional development is a reflective process that occurs during teaching and learning. Measuring competence requires assessing characteristics such as ideas, knowledge, skills, beliefs, and reflection (*Romijn, Slot & Leseman, 2021*). Research methodology typically favors quantitative studies, with relatively few qualitative and mixed methods studies available.

## Self-efficacy and professional development in digitalization

Participating in professional development for physical education teachers is a highly effective method for improving self-efficacy. The statistical analysis of the data reveals that self-efficacy is comprised of nine items, including overall teacher self-efficacy (*Ben Amotz et al., 2022*; *Gobbi et al., 2021*; *Koh et al., 2022*; *Kwon & Block, 2017*), computer self-efficacy beliefs (*Kalemoğlu Varol, 2014*), and digital self-efficacy (*Nguyễn, Đỗ & Nguyễn, 2022*; *Şahin, 2021*). These nine items were present in 90% or more of all cases of self-efficacy. Work engagement (*Gobbi et al., 2021*), perceived digital technology (*Maltagliati et al., 2021*), motivation (*Maltagliati et al., 2021*), stress (*Ben Amotz et al., 2022*; *Maltagliati et al., 2021*), and physical health (*Ben Amotz et al., 2022*) were also some of the factors associated with self-efficacy. A study analyzed the correlation between self-efficacy and digital technology among prospective physical education teachers. The study showed that there was a significant difference ($p < 0.05$), and also demonstrated that self-efficacy beliefs moderately affect attitudes towards digital technology (11% explanation rate) (*Kalemoğlu Varol, 2014*). Similarly, *Koh et al. (2022)* found that self-efficacy affects digital technology. Another study revealed that physical education teachers with technological prowess exhibited high levels of self-efficacy (*Şahin, 2021*), while those who struggled with technology had low self-efficacy. A study from Israel focused on the relationship between gender, distance learning, school policy support, and self-efficacy (F-ratio: 68.81; $p < 0.001$) (*Ben Amotz et al., 2022*). Female teachers had greater self-efficacy (adjusted $R^2 = 0.39$).

Contrary to the findings of this study, *Kalemoğlu Varol (2014)* suggested that male teachers may have a greater sense of self-efficacy in implementing digital technologies.

### Burnout and professional development in digitalization

Burnout is closely related to self-efficacy. The results indicated that four items were associated with burnout (*Ben Amotz et al., 2022*; *Maltagliati et al., 2021*; *Nguyễn, Đỗ & Nguyễn, 2022*; *Şirin et al., 2022*). The four items were present in 40% of all cases of burnout. In Vietnam, *Nguyễn, Đỗ & Nguyễn (2022)* examined the correlation between digital self-efficacy and burnout, showing that these two items have a correlated effect on professional development. It is remarkable that almost every project addressed the professional development presented. For instance, in distance learning, as a result of the higher self-efficacy of female teachers, female teachers can become more efficient (*Ben Amotz et al., 2022*). Each project explained the impact of professional development on self-efficacy and burnout during the discussion session. In contrast, we aim to explore the effect of self-efficacy and burnout on digital competence, and the professional development of physical education teachers. The literature has mentioned that emotional control, behaviors promoting physical activity, and intention to promote physical activity have made a difference in self-efficacy (*Maltagliati et al., 2021*), which in turn reduced stress and enhanced professional development intentions (*Ben Amotz et al., 2022*; *Gobbi et al., 2021*; *Koh et al., 2022*; *Kwon & Block, 2017*; *Nguyễn, Đỗ & Nguyễn, 2022*; *Şahin, 2021*; *Werner, 2020*). According to the Theory of Planned Action (*Ajzen, 1991*), intentions represent a person's commitment to engage in a specific behavior, exerting direct influence on a person's actions. According to *Maltagliati et al. (2021)*, for behaviors that promote physical activity, the intention is the physical education teacher's willingness to encourage a student's participation in physical activity. However, behaviors promoting physical activity are dependent on whether students are actively engaging in the exercise (*Maltagliati et al., 2021*). As a result, promoting intentions and behaviors in physical activity are influential factors in improving self-efficacy. In summary, self-efficacy has the highest impact on professional development. Burnout is another significant contributing factor to professional development, with stress being a good indicator of burnout.

In summary, severe burnout has impacted the emotions, self-efficacy, and teaching abilities of physical education teachers. Although professional development for physical education teachers is a process that involves training, conferences, and increasing teacher competence, the quality of professional development can be limited in the process by emotions and behaviors. Consequently, burnout can negatively impact the effectiveness of professional development.

## DISCUSSION

The ten articles selected for this review include physical education teachers, educational categories, digital tools, research methods and domains, main findings and discussion. According to yearly trends, researchers studied physical education teachers' digital self-efficacy and burnout levels peaked during COVID-19. Before COVID-19, research into the digital self-efficacy and burnout of physical education teachers had often been

overlooked. This review aims to analyze the effects of digital technology, self-efficacy, and burnout on the professional development of physical education teachers, as well as discuss future challenges in this area.

## Role of digital technology in professional development

Digital technologies have profoundly changed the pattern of professional development for PE teachers, providing new opportunities and challenges for education. Examples of this include physical education resources, virtual and augmented reality, pedagogical tools and apps (*Song et al., 2022*), data analysis and testing (*Ma & Zhou, 2021*), professional development and social media. Of these models, physical education resources dominate and include distance learning, online teaching, digital technology (*Sheng, 2020*), and blended learning (*Maltagliati et al., 2021*; *Wang et al., 2022*). Educators can use these technologies, such as simulating various sports arenas, scenarios, and motor skills training, to help students understand and practice. Consistent with research by *Koh et al. (2022)*, pupils utilize digital technologies (*e.g.*, Google Classroom and video software) to connect with their teachers and better understand exact movements through more intuitive slow-motion breakdowns.

Digital technologies provide PE teachers with more teaching tools and resources to improve the efficiency and quality of teaching and learning and broaden their professional development opportunities (*Brevik et al., 2019*). Not all PE teachers possess proficiency in digital technology, and they need professional development training to enhance their digital skills.

## Effect of self-efficacy on professional development

Italian researchers demonstrated that a physical education teacher's self-efficacy was linked to better digital professional development skills. This validates the idea that enhancing self-efficacy remains one of the most effective ways to facilitate teachers' professional growth (*Gobbi et al., 2021*). The research also stated that alterations in the educational environment and the academic setting affect the level of self-efficacy, which in turn affects professional advancement (*Şahin, 2021*). Similar to research findings by *Werner (2020)*, self-efficacy of the integration of technology in physical education is influenced by active mastery experiences and the feedback they receive during the learning and teaching processes. Active mastery skills are improved through professional development training, including conferences, online courses, and classroom sessions (*Werner, 2020*). The crucial role self-efficacy plays is in incentivizing individuals to participate in such opportunities.

However, the study showed that increasing professional development and policy support may be beneficial in promoting self-efficacy among physical education teachers, which is consistent with previous research (*Bond, 2021*). While quantitative methods explored and predicted the relationship between these variables, experiments tested the veracity of this relationship between variables. For example, *Kwon & Block (2017)* showed that after training in an adaptive physical education program, changes in self-efficacy differed between groups (*i.e.*, e-learning, traditional and control groups). The differences

were seen in the pre-test and post-test scores and the effect was significant. (F (1, 71) = 23.438, $p < 0.001$, partial eta-squared = 0.581.) This showed that professional development also affects the level of self-efficacy.

## Impact of burnout on professional development

One finding suggests that a lack of digital literacy leads to negative emotions (*Ben Amotz et al., 2022*), and that increasing opportunities for professional development can reduce these negative emotions. The scientific specialization of teachers relates to their level of psychological resilience (*Brouskeli, Kaltsi & Maria, 2018*). Another study suggested that "the maintenance of positive emotions can be used as a lever to promote insight into self-efficacy" (*Maltagliati et al., 2021*). Like the emotional exhaustion component of burnout (*Maslach, Schaufeli & Leiter, 2001*), charged emotions create a formidable barrier to the level of burnout. Consistent with previous findings, there was a significant correlation between burnout and self-efficacy (*Weißenfels, Klopp & Perels, 2022*).

Another study revealed that physical education teachers experienced moderate perceptual fatigue and digital burnout ($p < 0.05$). They reported no apparent feelings of pressure while teaching digitally (*Şirin et al., 2022*). However, when educators incorporated digital technology into their daily teaching routines, a significant correlation emerged between screen exposure time and the prevalence of perceptual fatigue and burnout (*Şirin et al., 2022*). These findings align with previous research indicating that full online teaching leads to severe burnout for teachers (*Nguyễn, Đỗ & Nguyễn, 2022*). In contrast, a hybrid teaching approach which utilises virtual reality (VR) and three-dimensional (3D) technologies has been shown to decrease levels of burnout. The literature suggests that different levels of burnout may be attributed to insufficient training in digital technologies before and after COVID-19 (*Nguyễn, Đỗ & Nguyễn, 2022*).

## Future professional development challenges

School support, digital technology education, emotions, and work engagement are all challenges for the future professional development of PE teachers. For example, school policy and leader support will motivate physical education teachers' active participation (*Aldosiry, 2022*), and young teachers' engagement in the work (*Lipponen, Hirvensalo & Salin, 2022*). Lifestyle challenges are also important factors. The mission of PE teachers is to motivate students to foster a healthy lifestyle (*Leeder & Beaumont, 2021*). It is worth noting how we can promote students' motor skills improvement through digital technology, drive the atmosphere of school sports, and continuously improve students' physical fitness, which are necessary for the professional development of physical education teachers.

## Limitation

This review has some limitations. First, it does not discuss the current situation of PE lectures using digital or mixed-mode teaching after COVID-19, despite the rapid development of digitalization during the pandemic. Second, VR and 3D technologies are not widely used in sports, particularly in education due to equipment limitations, and are

typically only implemented during sporting events. Consequently, this review cannot provide a comprehensive analysis of the role of VR technologies in professional development. Finally, there has been more focus on the link between self-efficacy and digital technology for PE teachers, rather than a focus on burnout and digital technology. The impact of digital technology on burnout is an underexplored area. Findings related to the professional development of physical education teachers are based on cross-sectional studies with limited available literature, which may pose certain constraints on their validation. Future research could explore experimental studies on digital PE teaching, its impact on teachers' professional development, and other empirical approaches.

## CONCLUSIONS

Self-efficacy and burnout are crucial factors in the professional growth of physical education teachers. The findings further validate that increased self-efficacy and reduced burnout positively influence the professional development of physical education teachers. This serves as a guide for the future directions of teaching. Enhancing self-efficacy and mitigating burnout are crucial objectives for the future professional development of PE teachers. Physical education teachers should prioritize improving self-efficacy and reducing negative emotions, such as stress, frustration, and anxiety. Educational institutions can support these goals by providing resources and training, and by fostering a positive work environment. Administrators should foster conditions that motivate teachers, thus helping them overcome burnout. This collaborative effort will result in increased professional development for physical education teachers and improved academic achievement for students. Educational institutions should prioritize the professional development of physical education teachers, cultivate and sustain a strong sense of self-efficacy, and implement measures to prevent and alleviate burnout. These institutions must recognize the importance of such actions to ensure the well-being and success of their physical education programs.

### Funding
The authors received no funding for this work.

### Competing Interests
The authors declare that they have no competing interests.

### Author Contributions
- Luhong Ma conceived and designed the experiments, performed the experiments, analyzed the data, prepared figures and/or tables, and approved the final draft.
- Chen Soon Chee analyzed the data, prepared figures and/or tables, authored or reviewed drafts of the article, and approved the final draft.
- Saidon Amri conceived and designed the experiments, analyzed the data, authored or reviewed drafts of the article, and approved the final draft.

- Xuejiao Gao performed the experiments, prepared figures and/or tables, authored or reviewed drafts of the article, and approved the final draft.
- Qinglei Wang analyzed the data, authored or reviewed drafts of the article, and approved the final draft.
- Nina Wang conceived and designed the experiments, prepared figures and/or tables, and approved the final draft.
- Pan Liu conceived and designed the experiments, performed the experiments, analyzed the data, authored or reviewed drafts of the article, and approved the final draft.

## Data Availability

This is a systematic review/meta-analysis.

## Supplemental Information

Supplemental information for this article can be found online at http://dx.doi.org/10.7717/peerj.18952#supplemental-information.

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
