# Peer review of "Impact of self-efficacy and burnout on professional development of physical education teachers in the digital age: a systematic review"

_PeerJ, doi:10.7717/peerj.18952_

## Round 0.1 · original submission · Minor Revisions

Thank you for your submission. The reviewers have identified a number of concerns that must be addressed.

**Language Note:** The review process has identified that the English language must be improved. PeerJ can provide language editing services - please contact us at [email protected] for pricing (be sure to provide your manuscript number and title). Alternatively, you should make your own arrangements to improve the language quality and provide details in your response letter. – PeerJ Staff

Reviewer 1 ·

Basic reporting

Writing could be improved to include more clarity in specific areas and structure of flow.
For example,
line 49-52: unclear transition statement. What do you mean by blended sport education? Expand and define, as sport education is also a pedagogical model. Be clearer in your point for this sentence.
line 56- "However, they" Who is they?
line 65-66: awkward sentence because it has not been addressed that this is a review, nor has the purpose been fully identified. This sentence disrupts the flow of the paper.
line 90 - "In conclusion" -delete, as this is not the conclusion..it is a summary of this piece
line 102- a systematic review of what? Be more specific here
line 105- make it clear that this is burnout in physical education, if this is the case.
line 111- What are primary digital tools for? Is it PE professional development or PE instruction? Be more clear about what you are referring to.
line 136- Were any data bases excluded? Why just these? Explain.
line 149- explain/expand on what Rayyan tool is and you need to cite it.
line 165- Explain and cite AXIS tools. How was this used, too?
line 179- Identify that these results were found in the extraction process
line 184-185 - use parentheses to sort out the number of articles for each type
line 196-197 - incomplete sentence and missing a closed parenthesis
line 208- three (articles) scored...
line 248- teaching models for X... be more specific...(Digital teaching models?)
line 271-272- In conclusion (summary, but not a conclusion)
line 295- define - explain more clearly what is meant by activity facilitating behavior and activity facilitating intentions
line 344- highly effective method for what?
line 392- double check how the word significant is used...sounds awkward
line 396- write out VR for the first time for those who do not know what VR means
line 405- hurt = negatively impact??? maybe use different language other than hurt
line 436- negative emotions towards what?
The writing in this manuscript improves from line 344 onward. Is there another writer here? It is obvious.

Experimental design

Design is appropriate, but could be explained more clearly. The steps get a little confusing. Is there a way you could outline the steps more clearly? Some of the tools need to be explained in more depth, as requested above (Rayyan and AXIS). Also, what data bases were excluded and why?

Validity of the findings

This is an interesting and useful study. The information is timely do to the increase demand of technology and also exploring the feasibility of technology in PE, especially since the pandemic. However, more clarity is needed in the description of the experimental design to make assumptions about the validity of the findings.

·

Basic reporting

From the perspective of the subject matter addressed, I can say that the research is interesting and relevant regarding the literature review on digital self-efficacy and burnout among physical education teachers.

However, I believe that the paper has some shortcomings, which I would like to mention further:


1 - There is some lack of clarity in presenting the ideas and the exact direction of the study. Each of the psychological constructs discussed in the article (self-efficacy, burnout, relationships between physical education teaching and digital technologies) has considerable literature.
The authors could offer a clear definition of key concepts used in the study, such as “burnout” and “digital self-efficacy”.
Furthermore, I believe the introduction should be divided into distinct paragraphs, each addressing
a specific aspect of the subject.
2 - The English language should be improved to ensure that an international audience can clearly understand your text. I suggest you have a colleague who is proficient in English and familiar with the subject matter review your manuscript, or contact a professional editing service.
3 - The year distribution of the studies is clearly explained in paragraph 3.3 (lines 213-222), so a figure seems to be redundant. I suggest removing the Figure 3.

Experimental design

4 - The original question in the introduction could be reduced. Questions number 1 (line 110), 4 (line 113), and 5 (line 115) are well-defined, relevant, and meaningful. I suggest removing or reformulating questions number 2 and 3 (lines 111-112) to narrow down the focus with respect to the topic and the results obtained.

Validity of the findings

5 - Conclusions should be linked better to the original research questions.

---

## Round 0.2 · Major Revisions

The previous Academic Editor is unavailable so I have taken over handling this submission in my capacity as Section Editor.

Please address Reviewer 3’s concerns in a revised submission.

Reviewer 3 ·

Basic reporting

GENERAL COMMENTS

The aim of this paper was to review the literature regarding the influence of self-efficacy and burnout on professional development of physical education teachers in the digital age. Although this article addresses an interesting topic, many issues should be addressed before publication.

SPECIFIC COMMENTS

INTRODUCTION
The introduction needs major revision and clarification. First, the aim of the study is not clear. Review papers should answer a specific research question, which is lacking in this study. Also, the way the authors build up their introduction does not lead to the research question. Although many of the necessary information regarding the background is not concise and, the authors should re-structure their introduction, explaining why this review is important. The authors described some studies on self-efficacy and burnout separately, but the discussion of both together is very shallow. Thus, it is recommend that the authors expand this part: how the self-efficacy and burnout will influence the professional development of physical education teachers. More importantly, this should lead to a clear research question.

METHODS
The methods section needs major revision. As it stands, it is not possible to replicate their study. The authors should justify why they have only searched papers from? The year should be more specific like from which months to which month? Why only from 2007-2023? The authors should also add what information they extracted as well as how they analyze the data more clearly. It seems that the authors did not screen the abstracts? After screening the title, they have looked at the full article. A quick search found 22 articles. I could not follow the method nor the flow.
Suggest to use subheadings to clearly show the information.

RESULTS
The results section needs major revision. The authors should add how they gathered all the information. Probably, including a research question would help the authors to structure their results. I think the authors put too many information in the tables, making the entire manuscript hard to follow. Table 1 check list need more clarification.

DISCUSSION
In the discussion section, the authors should further discuss their findings and the implication of these findings. They should also discuss their findings in more depth. However, in this section, the authors present many new results. These results should be moved to the results section. The authors also discuss many topics that are not related to the results.
Perhaps, part of this information should be moved to the Introduction section. In addition, they describe many studies in great detail which is not necessary for the discussion. This makes the discussion difficult to follow. The limitation is also not well thought. Please revise.

Thank you.

Experimental design

Research question is not well defined and relevant. Need revision.
A quick search found 22 articles, but authors only found 10 articles.

Validity of the findings

RESULTS
The results section needs major revision. The authors should add how they gathered all the information. Probably, including a research question would help the authors to structure their results. I think the authors put too many information in the tables, making the entire manuscript hard to follow. Table 1 check list need more clarification.

DISCUSSION
In the discussion section, the authors should further discuss their findings and the implication of these findings. They should also discuss their findings in more depth. However, in this section, the authors present many new results. These results should be moved to the results section. The authors also discuss many topics that are not related to the results.
Perhaps, part of this information should be moved to the Introduction section. In addition, they describe many studies in great detail which is not necessary for the discussion. This makes the discussion difficult to follow. The limitation is also not well thought. Please revise.

Additional comments

Unfortunately, the manuscript requires major attention.

---

## Round 0.3 · accepted · Accept

Congratulations, your revised manuscript has been approved for publication

·

Basic reporting

No comment

Experimental design

No comment

Validity of the findings

No comment

Additional comments

Dear Authors,
My opinion on the paper is that you actively responded to reviewers' comments.
I appreciate the latest version of the manuscript.
Yours sincerely

Reviewer 3 ·

Basic reporting

I believed the authors are utilising a search focused on Chinese's engine, which may explain why the number of available articles is lower in comparison to databases such as the Web of Science (WOS) and Scopus. Despite this limitation, the authors have successfully addressed all the comments and feedback provided during the review process. Their thorough revisions demonstrate a commitment to enhancing the manuscript's quality and relevance. Therefore, I propose accepting this manuscript for publication. Thank you for your consideration.

Experimental design

A lot more understandable. However, I propose sending the manuscript for native English proofreading.

Validity of the findings

Yes. Acceptable now.

Additional comments

I propose accepting this manuscript for publication. Thank you for your consideration.